# Exploring Conditions for Handling Packing and Shipping *Aedes aegypti* Males to Support an SIT Field Project in Brazil

**DOI:** 10.3390/insects13100871

**Published:** 2022-09-25

**Authors:** Maylen Gómez, Aline T. Macedo, Michelle C. Pedrosa, Fernanda Hohana, Verenna Barros, Bianca Pires, Lucas Barbosa, Miriam Brito, Luiza Garziera, Rafael Argilés-Herrero, Jair F. Virginio, Danilo O. Carvalho

**Affiliations:** 1Biofábrica Moscamed Brasil, Quadra D-13, Lote 15, Distrito Industrial do São Francisco, Juazeiro 48909-733, Brazil; 2Insect Pest Control Subprogramme, Joint FAO/IAEA Programme of Nuclear Techniques in Food and Agriculture, P.O. Box 100 Vienna, Austria

**Keywords:** sterile mosquitoes, shipment, knockout, packaging densities, chilling, quality control

## Abstract

**Simple Summary:**

*Aedes*-transmitted diseases are one of the most burdensome problems for public health worldwide. These diseases can be controlled by reducing the vector population, which can use alternative methods by releasing sterile male mosquitoes. Here, we show the impact of chilling, compacting, and transporting procedures on male quality. Our results reveal the temperature required for the absolute knockout of the *Aedes* males from a local strain. We recorded an adverse effect of chilling on the escape rate; however, after 24 h of recovery, we did not detect it. The interaction between density and transportation time impacted the survival, insemination, and flight ability rates. These outcomes are essential to developing *Aedes* packing and shipping protocols for the mosquito sterile release technique.

**Abstract:**

The sterile insect technique (SIT) application, as an alternative tool for conventional mosquito control methods, has recently gained prominence. Nevertheless, some SIT components require further development, such as protocols under large-scale conditions, focusing on packing and shipping mosquitoes, and considering transporting time. Immobilization of *Aedes aegypti* males was tested at temperatures 4, 7, 10, and 14 °C, and each temperature was assessed for 60, 90, and 120 min. The recovery after 24 h was also studied. Chilled and control-reared males had comparable survival rates for all conditions, although 4 °C for 120 min impacted male survival. The male escape rate was affected after 60 min of exposure at 4 °C; this difference was not significant, with 24 h of recovery. First, we defined the successful immobilization at 4 °C for 60 min, thus enabling the evaluation of two transportation intervals: 6 and 24 h, with the assessment of different compaction densities of 100 and 150 mosquitoes/cm^3^ at 10 °C to optimize the shipment. Compaction during simulated mosquito shipments reduced survival rates significantly after 6 and 24 h. In the mating propensity and insemination experiments, the sterile males managed to inseminate 40 to 66% for all treatments in laboratory conditions. The male insemination propensity was affected only by the highest compaction condition concerning the control. The analysis of the densities (100 and 150 males/cm^3^) showed that a higher density combined with an extended shipment period (24 h) negatively impacted the percentage of inseminated females. The results are very helpful in developing and improving the SIT packing and shipment protocols. Further studies are required to evaluate all combined parameters’ synergetic effects that can combine irradiation to assess sexual competitiveness when sterile males are released into the field.

## 1. Introduction

Arbovirus diseases such as chikungunya, dengue, and Zika have become urgent and are present in many countries, with a continually increasing number of cases yearly [1,2]. For example, the most significant number of dengue cases reported globally was in 2019, in which the American region alone reported 3.1 million cases during this year, with more than 25,000 severe cases. According to the Pan-American Health Organization (PAHO), in the region of the Americas, a total of 2.3 million cases of Dengue were reported in 2020. Of those, 63% were only notified in Brazil (1.5 million cases), including 828 severe cases and 554 deaths [2,3,4]. In addition, Zika, a neglected disease, reappeared in 2015, causing global concern and resulting in congenital malformations, microcephaly, and fetal death. Between 2015 and 2016, the Zika virus increased the number of countries reporting this disease from 10 to more than 80, with more than 1.9 million cases in the American continent. Brazil is the most affected in the Americas, representing more than 80% of the cases consistently [4,5].

In Brazil, both diseases have a common vector, the female mosquito of *Aedes aegypti*, widespread in urbanized areas, highly anthropophilic, and well adapted to artificial breeding sites, without a specific predator. The standard practices to control the disease are related to the patient’s drug treatment and, holding the mosquito population, avoiding virus dissemination [6,7]. The vector control usually comprises three subdivisions, biological, mechanical, and chemical. Each contains a set of activities that, when used in different levels and combinations, can provide the best tools that can be implemented for a specific area, targeting an area-wide approach to reduce transmission [8,9,10]. Besides the Brazilian efforts to control these arboviruses through a national plan to prevent them, the current activities are not enough to prevent people from getting infected [11,12].

One crucial factor is the constant pressure of selecting mosquito populations resistant to insecticides, reducing the number of substance options, or increasing the amount used significantly [13,14,15]. In the last decade, Brazil has initiated a new chapter, willing to evaluate new approaches to controlling the mosquito population using alternatives [16,17], such as genetically modified organisms for population suppression [18,19]; mosquito population replacement carrying a symbiont [20,21,22]; the use pyriproxyfen in dissemination stations increasing juvenile mortality [23,24]; and at last, the use of sterile males, as proposed in this study [16,25].

The sterile insect technique (SIT) effectively and efficiently controls insect populations considered pests for agriculture and livestock, counting several successful programs worldwide [26]. The SIT application is an innovative and sustainable alternative for *Ae.*
*aegypti* mosquito control recently has gained significant visibility [27,28], with several field pilot projects implemented in the last few years showing the SIT’s effectiveness in reducing target mosquito populations in countries such as Cuba, Italy, Spain, and the USA [29,30,31]. The technique comprises the mass-rearing of the species of interest, followed by sterilization with ionizing irradiation, the release of sterile males in the target area, and keeping up with the population suppression adjusting parameters to achieve the suppression [32]. In most scenarios, the target area can be far away from the male production facility or radiation source, thus requiring the shipment of chilled and compacted (sterile) males over long distances. For that, efficient pre-release protocols for packing and shipping these sterile mosquitoes are crucial to easily handling them without compromising their quality and performance in the field. Recent scientific pieces of evidence already show that low temperatures for immobilizing sterile mosquitoes affect behavioral parameters, such as insemination rate, survival, and mating competitiveness [33,34,35]. Additionally, compaction causes physical damage to wings, antennas, and other body parts, which could negatively affect the dispersal and/or the sexual performance of sterile males in the field [36,37].

Currently, the effects of chilling, compaction, and transport at low temperatures on male quality are not yet well understood and only tested in specific situations [35,36,37,38,39,40]. Considering this scenario, we studied the effects of chilling, compaction, and shipment interval to develop a protocol targeting non-irradiated male transportation over long distances. This study focused on a specific pilot trial conducted in Brazil, addressing local conditions and needs to fulfill three main objectives: (1) to determine the optimal temperature and incubation period for mosquito knockout; (2) to determine the handling and packing conditions during the transport of male mosquitoes for long distances; and (3) to assess the chilling, packing, and shipping impact on male quality.

## 2. Materials and Methods

### 2.1. Mosquito Colony and Rearing Conditions

The *Ae. aegypti* strain was established from eggs collected in several neighborhoods in Recife in 2017. Since 2018, this local strain has been reared using standard climate-controlled conditions [27 ± 1 °C; 70 ± 10% RH; and photoperiod of 12:12 (L:D) hours] following the BMB protocol described by Carvalho et al. (2014) [38]. All experiments were conducted with non-irradiated mosquitoes.

### 2.2. Conditions for Mosquito Chilling

For this study, handling’s knockout definition should indicate the temperature and time in which all mosquitoes fall off the container walls. The candidates’ temperature and time used the studies from Culbert et al. and Chung et al. as starting points, as well as their results as a reference point [39,40]. Males were counted to compose seven groups of 50 individuals and placed inside containers (500 mL; 7.65 cm height and 12 cm diameter). The pots containing adult males adults were transferred to a vertical incubator (SPLABOR model: SP-227/364 temperature range of ±0.5 °C, SP Labor, Pres. Prudente, Brazil) for immobilization at different temperatures of 4, 7, 10, and 14 °C, and each temperature was held for 60, 90, and 120 min. The incubator is located inside the insectary to avoid great temperature differences. After each group’s immobilization, we transferred the mosquitoes to cages for 60 min of recovery with 10% sucrose *ad libitum*. After that, we determined the immediate mortality by counting dead insects found after recovery. Mortality was checked daily for the next 30 days to determine its survival (item 2.4); we also assessed the escape rates (2.5) of insects exposed to knockout conditions. The control group was defined as mosquitoes from the same batch and age but not chilled.

### 2.3. Conditions for Mosquito Packing during Shipping

A subsequent study aimed to define suitable shipment conditions for mosquitoes, but we focused on our pilot site’s logistics using the best knockout condition found in the previous study. After the knockout at 4 °C for 60 min (using the same procedure described in point 2.2), mosquitoes were packed into plexiglass cubes (2.2 × 2.2 × 2.2 cm—10.6 cm^3^) in two different densities, 100 and 150 mosquitoes/cm^3^. They were selected based on the less impacting density found by Culbert et al. [36]. The compacted mosquitoes were placed inside a prechilled cooler (Dometic CFX 50—Dometic, Solna, Sweden—with an internal capacity of 580 mm × 385 mm × 360 mm at 10 °C) and transferred to a cold chamber (2.3 m × 2.3 m—Genfrio Refrigeration Commerce and Service Ltd., Recife, Brazil with a temperature range of ±1 °C) also set at 10 °C, in which each density was kept for 6 and 24 h, simulating our pilot trial’s minimum and maximum shipping intervals. This temperature was chosen considering the results of the insemination and flight ability rates reported by Culbert et al. [41]. At the end of the transport simulation interval (also referred to as “shipping” or “shipment” along the manuscript), insects were transferred back to the insectary using the same cooler and conditions. At the insectary, the males were transferred into cages to recover for 60 min with 10% sucrose and water solution *ad libitum*. The distance between the insectary and the cold chamber is short, less than 1 min walk, from one door to the other at the same corridor. We evaluated the physical damage, survival, insemination, and escape rate besides the immediate mortality. For all experimental groups, we used mosquitoes from the same batch, and they were defined as (1) control group (not chilled, nor compacted and without shipment); (2) chilled (chilled at 4 °C and 60 min, but not compacted nor shipped); (3) 100 cm^3^ and 6 h exposure, (4) 100 cm^3^ and 24 h exposure; (5) 150 cm^3^ and 6 h exposure; and (6) 150 cm^3^ and 24 h exposure.

### 2.4. Survival Analysis

Each condition’s survival had batches with 50 males (with seven repetitions) carefully aspirated from the recovery cages and placed in containers (500 mL; 7.65 cm height and 12 cm diameter) for daily observation for the following 30 days. During this period, males had access to 10% sucrose *ad libitum*. Dead mosquitoes were removed daily, and the event of death was recorded. After 30 days, the remaining mosquitoes were recorded as censored, and a survival curve from a Kaplan–Meier log-rank analysis was performed.

### 2.5. Escape Rate

The flight ability test uses the device developed by the Insect Pest Control Section of the Joint FAO/IAEA Centre of Nuclear Techniques in Food and Agriculture to determine the escape rate [41]. Briefly, after exposing mosquitoes to packing and shipping conditions, 100 males from each group (with 4 repetitions), including a control, were accommodated in the appropriate lower chamber, according to the protocol, initiating the evaluation, which lasted for 2 h. The number of fliers and non-fliers was recorded at the end, and the escape rate was defined as the percentage of fliers from the total insects at the beginning of the experiment. The control group consisted of non-chilled males from the same group/rearing batch and generation. The escape rate of mosquitoes from item 2.2 was performed one and 24 h after the chilling exposure.

### 2.6. Physical Damage

The insect cage was kept at 0 °C overnight for physical damage assessment to kill the insects without further damage using all insects after removing the samples for survival and the escape rate assessment (for 100 cm^3^ with 410 to 1295 mosquitoes per cage, while 150 cm^3^ had 728 to 1091). After that, insects were examined under a stereo microscope (Leica, model EZ4D, Wetzlar, Germany, at 40× magnification) to check the percentage of insects, which records any damage on wings (broken), legs (at least one broken and missing), and a visual qualitative scale loss (estimating around 70% loss of body scales) over the body from only those treatments using 100 and 150 males/cm^3^ since control groups would not present physical damage.

### 2.7. Mating Propensity and Insemination

To assess mating propensity, we followed Balestrino et al. with modifications [42]. Briefly, we determined the number of inseminated females after 15 h caged with a single male from the treatments tested. The number of females for this experiment was limited to 10 individuals. For that, a single virgin male (around 72 h after emergence) was placed with ten receptive virgin females in a plastic cage (30 cm × 30 cm × 30 cm) for 15 h (comprising two twilight periods) with continuous access to a 10% sucrose solution. The period was selected based on the time spent during the coitus [43,44] and the male latency period as an indicator of the mating competitiveness vigor [45,46,47]. Thus, we dissected each female under a stereomicroscope (Leica, model EZ4D, Wetzlar, Germany) to remove their spermathecal capsules for evaluation. We evaluated each female’s status (presence or absence of sperm) and level of insemination (partially or fully). We considered a female partially inseminated when at least one of their spermathecae had sperm while a full inseminated when at least two spermathecae contained sperm. Each group of treated males had eight replicates.

### 2.8. Statistical Analysis

The quantile-quantile plot observation checked the data normality in combination with the data frequency distribution. As they were not normally distributed, all research was performed using non-parametric tests, such as Kruskal–Wallis and Wilcoxon–Mann–Whitney tests, using an alpha level of 0.05. The Kaplan–Meier method was used to generate the survival curves and the Log-Rank test to perform pairwise comparisons of different treatments. All statistical analyses were performed using R language using the RStudio environment [48,49]. The essential packages and version information can be found in the Appendix A.

## 3. Results

### 3.1. Impact of Chilling Procedure on Survival and Escape Rate

We observed no immediate mortality regardless of the duration time assessed at 4 and 14 °C. However, insect mortality happened at 7 and 10 °C, with a mortality rate lower than 1% and an average of 0.3% (±0.13%) for all three evaluated time intervals (Appendix A).

The survival curves of males exposed from 4, 7, 10, and 14 °C are presented in Appendix A. Survival rates reached values higher than 70% after 30 days for all evaluated chilling conditions. The males’ survival curves for temperatures 7, 10, and 14 °C showed no statistical difference among all chilling assessed periods. Results related to the male survival after exposure at the temperature of 4 °C showed a significant decrease in this behavioral parameter only after exposure to the highest chilling time of 120 min (χ^2^ = 10.6; df = 3; *p* = 0.01) (Figure 1A). The temperature required for the absolute knockout of the males was 4 °C. Temperatures between 7 and 14 °C caused partial immobilization of the insects, most of them hanging on the container’s walls. When observing chilled mosquito survival for 60 min, there was no significant difference between the four temperatures assessed (4, 7, 10, and 14 °C) and untreated males (χ^2^ = 5.6; df = 4; *p* = 0.231) (Figure 1B).

Based on the results of male survival curves after chilling at different temperatures, the male escape rate was assessed for mosquitoes exposed at four °C for 60 min with the statistical difference among all groups (Kruskal–Wallis = 8.371; df = 3; *p* = 0.039—Figure 2). Significant differences were seen in the escape rate between the control group and the males’ chilled group (control group: 69.6%; chilled adults: 54.5%) (Wilcoxon–Mann–Whitney = 35.5; *p* = 0.045, Figure 2). In contrast, no significant differences were found in the mean escape rate between untreated and chilled males twenty-four hours after chilling (control group: 70.9%; chilled adults: 70.4%) (Wilcoxon–Mann–Whitney = 24; *p* = 1, Figure 2). The control group’s escape rate was not impacted by age (Wilcoxon–Mann–Whitney = 24; *p* = 1.00), while in the exposed group, age affected this parameter (Wilcoxon–Mann–Whitney = 5; *p* = 0.022).

### 3.2. Impact of Packing and Shipping Procedures on Survival, Insemination, Escape Rate, and Physical Damage

The immediate mortality obtained after recovery pointed to a higher impact with the higher compaction density (Appendix A), showing that longer intervals seem not to have an effect. Compared to the control, all treatments had some impact on immediate mortality, ranging from 0.81% to 2.48%, with a mean of 1.29% (±1.2%), while the control had no immediate mortality (Appendix A).

During the first 15 days, the transporting conditions tested seem to have a low impact on survival rates, with survivorship above 80% for all treatments (Figure 3A,B). However, considering the whole observation period, the assessed density conditions reduced survival significantly when males were compacted for 6 and 24 h at 10 °C (6 h with χ^2^ = 135; df = 3; *p* < 0.05—Figure 3A, and 24 h χ^2^ = 232; df = 3; *p <* 0.05—Figure 3B). Survival curves were similar for shipment of 6 and 24 h with mosquitoes compacted at 100 males/cm^3^ (χ^2^ = 2.5; df = 1; *p* = 0.12, Figure 3C). On the other hand, there was a statistical difference observed in male survival at 150 males/cm^3^ compaction for 6 and 24 h. (χ^2^ = 10.2; df = 1; *p* < 0.001, Figure 3D).

The mean proportion of mated females ranged from 40 to 66% for all treatments. The 6 h shipment interval did not show statistical significance among all tested groups (Kruskal–Wallis = 7.00; df = 3; *p* > 0.05). At this 6 h interval, the male insemination propensity was affected only by the highest compaction condition in relation to the control (Wilcoxon–Mann–Whitney = 80.5; *p* = 0.019—Figure 4A). Regarding the shipment interval (6 h and 24 h) the 100/cm^3^ density showed no statistical difference (Wilcoxon–Mann–Whitney = 49.5; *p* = 0.152), as well as the 150/cm^3^ density (Wilcoxon–Mann–Whitney = 53; *p* = 0.075—Figure 4B). Males compacted at 100/cm^3^ for 06 h were able to inseminate a similar proportion as the one found in the untreated (Wilcoxon–Mann–Whitney = 64; *p* = 0.288—Figure 4A). The analysis of the densities (100 and 150 males/cm^3^) showed that a higher density independent of the shipment interval negatively impacted the percentage of inseminated females (Wilcoxon–Mann–Whitney = 209.5; *p* < 0.05).

The spermatheca replenishment level was determined as partially and fully inseminated rates of all inseminated females, as presented in Figure 5. For the 06 h shipment, the insemination pattern was shifted and had a statistical difference between the two densities assessed for partially inseminated females (Wilcoxon–Mann–Whitney = 84.5; *p* < 0.05, Figure 5). Interestingly, the 24 h shipments did not show this observed insemination pattern shift (Wilcoxon–Mann–Whitney = 36.5; *p* = 0.119—Figure 5).

The escape rate results indicated no statistical significance among all groups exposed to 6 h shipment interval at 10 °C (Kruskal–Wallis = 1.415; df = 2; *p* = 0.493—Figure 6A). In detail, males compacted in both densities, with a shipment interval of 6 h, had no significant impact compared to the control group (Wilcoxon–Mann–Whitney = 12; *p* = 0.394, Figure 6). Meanwhile, the 24 h interval showed a difference among all groups (Kruskal–Wallis = 12.29; df = 2; *p* < 0.05), this difference is also significant between the two tested densities (Wilcoxon–Mann–Whitney = 1; *p* < 0.05, Figure 6). However, males compacted at a density of 100 males/cm^3^ for 24 h had a significantly lower escape rate than untreated males and compacted males at 150 males/cm^3^ (Wilcoxon–Mann–Whitney = 36; *p* < 0.05, Figure 6). Moreover, the mosquito escape rate showed a statistical difference between the two shipment periods compacted at 100 males/cm^3^ density (Wilcoxon–Mann–Whitney = 31; *p* = 0.041, Figure 6), which males packed for 24 h had a lower escape rate. It is worth mentioning that the escape rate using 150 males/cm^3^ density had no significant difference between both simulated shipments (6 and 24 h) (Wilcoxon–Mann–Whitney = 22; *p* = 0.589, Figure 6).

Regarding the physical damages, less than 2% of mosquitoes had damaged their wings after exposure to both compaction densities for 6 and 24 h at 10 °C (Figure 7A) without a statistical difference found between the densities for each shipment interval. Observations lead to a higher wing damage percentage for both compaction densities in more extended shipment periods (100 males/cm^3^ with Wilcoxon–Mann–Whitney = 12.5; *p* = 0.015 and 150 males/cm^3^ with Wilcoxon–Mann–Whitney = 12; *p* = 0.013). Less than 1% of compacted mosquitoes had broken legs (Figure 7B), without any statistical difference between densities (100 males/cm^3^ with Wilcoxon–Mann–Whitney = 38.5; *p* = 0.930 and 150 males/cm^3^ with Wilcoxon–Mann–Whitney = 35; *p* = 0.690) or shipment periods (6 h with Wilcoxon–Mann–Whitney = 50; *p* = 0.410 and 24 h with Wilcoxon–Mann–Whitney = 46; *p* = 0.630). In contrast, the densities had an increased number of males with severe scale loss after 6- and 24-h exposure. There were no statistical differences observed between the shipment time (6 and 24 h) for each density (100 males/cm^3^ with Wilcoxon–Mann–Whitney = 49.5; *p* = 0.420 and 150 males/cm^3^ with Wilcoxon–Mann–Whitney = 38; *p* = 0.890). However, it was seen a substantial scale loss for 150 males/cm^3^ density, regardless of its shipment period (6 h with Wilcoxon–Mann–Whitney = 10; *p* < 0.05 and 24 h with Wilcoxon–Mann–Whitney = 16; *p* = 0.035, Figure 7C).

## 4. Discussion

The SIT’s success relies on the sterile male ability to compete effectively with wild males to mate with wild females in the target area. The SIT has proven successful against numerous pest insects in agriculture and livestock worldwide [50]. In the case of mosquitoes, the SIT package is still under development and refinement, which is a work in progress involving parameters related to handling, packing, irradiation, transportation, and release procedures [51,52,53]. Among these procedures, the chilling process can make it easy to handle and compact insects; this process can deeply impact their sexual competitiveness [54]. Consequently, impacting sexual competitiveness, the releases become practically ineffective, compromising the SIT program’s success [50,54]. Some behavioral parameters participate in the competitiveness of sterile males in the field, the assessment of parameters involved with mating and behavior before the releases are crucial since they reflect the male quality [55,56]. In the present study, we focused on evaluating the impact of different packing and shipping conditions on male quality parameters, such as the body structures’ physical damage, survivorship, escape rate, and mating propensity.

The results obtained from the chilling temperatures and the shipping intervals for mosquito knockout showed a low impact on the immediate mortality rate and survival over 30 days. All treatments exhibited a similar survival pattern compared with the control. Interestingly, our results indicated that only the lowest temperature (4 °C) ensured males’ absolute knockout. The period of 60 min ensures enough time not only for the mosquitoes to knockout but also to have time for proper handling of a large number of mosquitoes in an operational program. However, incubating sterile males at this temperature of 4 °C for at least 120 min presented a significant decrease in male survival (Figure 1A). For this same species, the study by Culbert et al. (2018) testing the following temperature at 0, 4, 8, 10, and 25 °C and holding those temperatures for 120 min, showed that the 0 °C was the only one with any impact on the male survival [36]. Similar results were found using the *Anopheles arabiensis* species by the same author [57]. Interestingly for *Aedes albopictus*, the effect on male survival incubated up to 3 h at 5 °C had no adverse impact compared with the control. Conversely, the male survival decreased at 10 °C when incubated for 60 min [37].

The escape rate assessment immediately carried out after mosquito knockout showed a significant decrease in male flight capability after incubation at 4 °C for 60 min (the control group had a 69.6% escape rate, while the knockout group had 54.5%) (Figure 2); however, when a similar exposed group had 24 h to recover from the exposure, we found that the males manage to have a similar escape rate as the control group (which was not chilled or recovered), with an escape rate of 70.4% and 70.9%, respectively. To assess whether it was related to the male age, we checked that there was no difference between the tested ages for the control males at 72 and 96 h. An alike hypothesis was brought by Shelly et al. (2010), finding a similar result evaluating the mating competitiveness of sterile *Ceratitis capitata* males after the chilling [54].

From all evaluated parameters for chilling, we found that the escape rate (provided by the flight ability test) was more sensitive than survival and immediate mortality to detect a potential impact after 60 min at 4 °C. The escape rate was pointed out as an excellent quality control parameter since it is sensitive to distinguish differences among species, ages, and stress levels (i.e., irradiation, chilling, compaction), which reflects directly the male quality [39,58]. During an operational program is essential to have quicker tools to provide data to help managers make timely decisions. In addition, the escape behavior of chilled males should be better studied and correlated with their accurate sexual competitiveness under laboratory and field conditions.

The packing and shipping results suggest that the compaction density significantly impacted the survival rate, which can be potentialized by shipping intervals (6 and 24 h). The increase in time using 100 mosquitoes/cm^3^ showed similar survival rates, while the compaction density of 150 mosquitoes/cm^3^ kept for 24 h reduced the male survival rate significantly. The study conducted by Chung et al. (2018) using 80 males/cm^3^ at 7 and 14 °C and incubated for 24 h showed that the survival rate (evaluated 24 h after exposure) of males was around 60% at these two temperatures, similar to the results found in our study at a density of 100 males/cm^3^ at 10 °C for 24 h [40]. Our findings indicated that density and transportation time are crucial in reducing the survival rate above 100 males/cm^3^.

Surprisingly, the highest compaction density did not impact the escape rate, regardless of the shipping period. The observed escape rate of *Ae. aegypti* males using 150 mosquitoes/cm^3^ was equal to untreated males for both shipment intervals (6 and 24 h). On the other hand, our results indicated a reduction in the escape rate using 100 mosquitoes/cm^3^ for 24 h. This finding could be the result that at this density, the males would have some more space among them in the container, which during the attempt to move, could increase their physical interaction, resulting in more physical damage. Although Figure 7 shows that less than 1.5% of the observed males had wing damage, around 2% showed leg damage and around 10% scale loss. With that in mind, we could still point out that the longest shipping interval led to slightly higher wing damage (Figure 7), and despite not being statistically different, the 100 males/cm^3^ density incubated for 24 h had the highest physical damage percentage, negatively impacting its escape rate, reaching less than 70%. Chung et al. (2018) also share the same hypothesis, observing more significant physical damage at lower densities [40]. This could be avoided in a more compact density due to the reduced confining space or a temperature lower than 10 °C to keep the mosquitoes in a complete knockout status. In both cases, further studies should be developed to explore the interacting balance of temperature, humidity, time, and compaction to verify this hypothesis [40].

The compacting densities and the shipping intervals had less than 1% of physical damages, specifically for wing and leg damage. However, a study conducted by Chung et al. (2018) showed that for wing damage, the densities of 240 males/cm^3^ had the highest damage percentage and the density of 40 males/cm^3^ with the most significant damage for the legs [40]. In contrast, regardless of the time exposure of 6 and 24 h, both densities had more than 90% of individuals with scale loss, and it is higher at 150 males/cm^3^ with 99.1% and 99.2% of individuals with scale loss between 6- and 24-h shipment interval. Chung et al. (2018) similarly observed a high scale loss (more than 90%) when mosquitoes were compacted at 40 and 240 males/cm^3^ density for around 20 h shipment [40]. In summary, density is a crucial factor for scale loss in mosquito *Ae. aegypti*, although some differences between the methods and strain origins could also be relevant to spot the differences between the studies.

The insemination rate was more impacted by the highest density (150 males/cm^3^—6 h shipment) with around 49% of females inseminated than by the lower density (100 males/cm^3^—6 h shipment) with 58%. When the shipping interval increased to 24 h, the female insemination percentage was impacted again for both densities (100 males/cm^3^ had 50%, while 150 males/cm^3^ had 40% insemination). Despite its reduction, it is relevant to mention that our findings statistically show that shipment interval did not impact the insemination rate of females. Similar to the findings of Culbert et al. (2018), which showed the insemination rate of *Ae. aegypti* to be less sensitive to the compaction, our results also did not show statistical difference among the compaction levels used, although the highest density (150 males/cm^3^) had a statistical difference when compared with non-compacted/chilled control (Figure 4) [41]. Finally, compaction did not affect the full insemination rate; this result contrasts with the study of Culbert et al. (2018) [41]. We also have to consider the role of different strains, species, insect age, and methodological performance in this response, as was recently pointed out by Maïga et al. (2022) [58].

Consistent with other studies, we found that chilling, packing, and conditions kept during the shipping (temperature and compaction) have an impact on male quality, resulting not only from adverse effects of temperature, compaction, and shipment duration but also from the interaction among these parameters [37,54,59]. The overall analysis, combining all parameters evaluated, shows that survival, insemination, and escape rate were able to detect adverse effects on male quality. The goal of a mosquito operational SIT program is to reduce the wild mosquito population by the over-flooding releases of competitive sterile males in the target area [50]. Thus, it is essential to ensure successful mating and transfer of sterile sperm after undergoing the three main processes: rearing, irradiation, and release. In this context, our results point out that survival, insemination, and escape rate have the potential to be considered as components of the quality control assessment to predict the male fitness and mating quality and bring this information timely to detect potential alterations during post-irradiation procedures before the release. In summary, the results are exceptionally helpful in developing and improving the SIT package protocols. Future studies must evaluate all combined parameters’ synergetic effects that could be combined with irradiation and assess the sexual competitiveness when released in the field.

## 5. Conclusions

The study’s main goal was to explore handling, packing, and shipping conditions *Ae. aegypti* males to a distant SIT pilot trial in Brazil. Our data consistently showed that to ensure an optimal chilling procedure, under our local situation, the *Ae. aegypti* males must be immobilized at 4 °C for 60 min, using a compaction density of 100 males/cm^3^ with a maximum transporting time of 6 h. The survival, insemination, and escape rate can be used synergistically to provide timely information about the negative impact of packing and shipment on male quality. These results were valuable in establishing a packing and shipment protocol for our pilot project focusing on our specific conditions and logistics. In addition, the proposed protocol can be improved for more extended shipment periods considering longer distances between our facility and potential release areas in Brazil.

## Figures and Tables

**Figure 1 insects-13-00871-f001:**
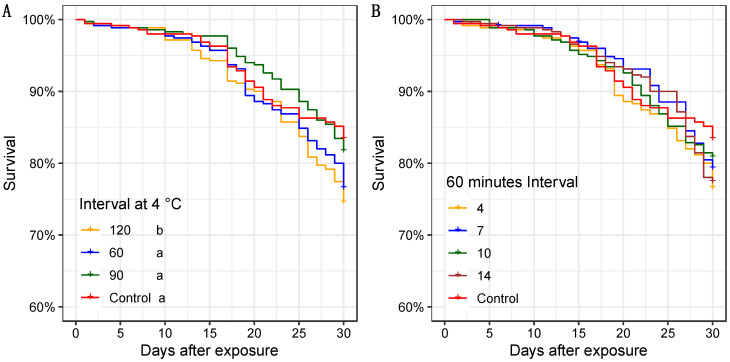
Kaplan–Meier survival curves of chilled males of *Ae. aegypti* after exposure to different chilling conditions for 30 days without compaction: (**A**) Chilling time affects male survival rates over time; (**B**) Temperature effect after 60 min of exposure on male survival rates over time. Different letters represent significant differences among the treatments for the time interval and the temperature (Log-rank test; *p* < 0.05 for (**A**) and *p* = 0.230 for (**B**)). All time intervals and all temperatures used mosquitoes from the same batch and shared one control group. T = 27 ± 1 °C; RH = 60 ± 10%, photoperiod = L12:D12.

**Figure 2 insects-13-00871-f002:**
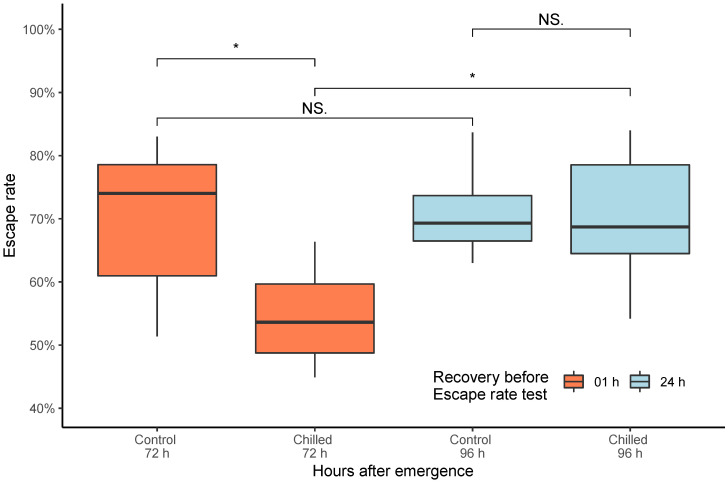
*Aedes* mosquito male escape rates after chilling at 4 °C for 60 min without compaction. Except for the controls (which only represent hours after emergence), the light blue group (96 h) was exposed to chilling at the same time as the 72 h, but they had 24 h to recover prior to the escape rate test. Significance values goes as ‘*’ < 0.05, ‘NS’ < 1. The boxplot shows the median and upper and lower quartiles.

**Figure 3 insects-13-00871-f003:**
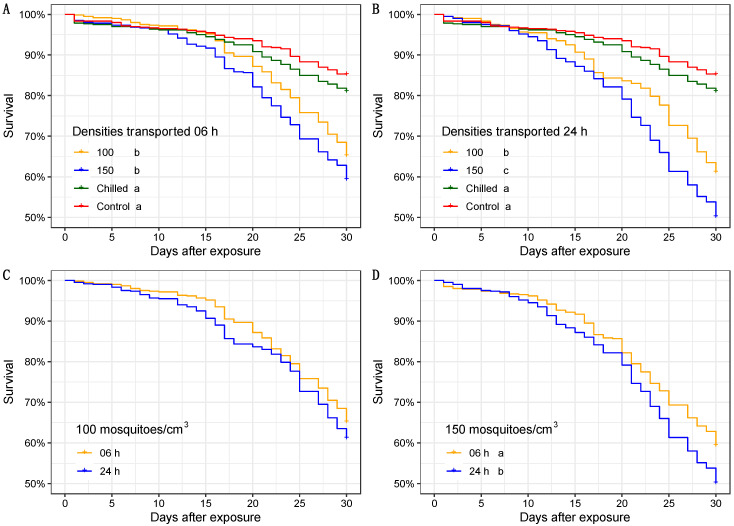
Kaplan–Meier survival curves of compacted and control males over 30 days: (**A**) Chilling and compacting effect for 6 h; (**B**) Chilling and compacting effect for 24 h; (**C**) Shipment using 100 males/cm^3^ after knockout (Log-rank test; *p* < 0.12); (**D**) Shipment using 150 males/cm^3^ after knockout. Differences between treatments in (**A**,**B**,**D**) are represented by different letters (Log-rank test; *p* < 0.05) T = 27 ± 1 °C; RH = 60 ± 10%, photoperiod = L12:D12.

**Figure 4 insects-13-00871-f004:**
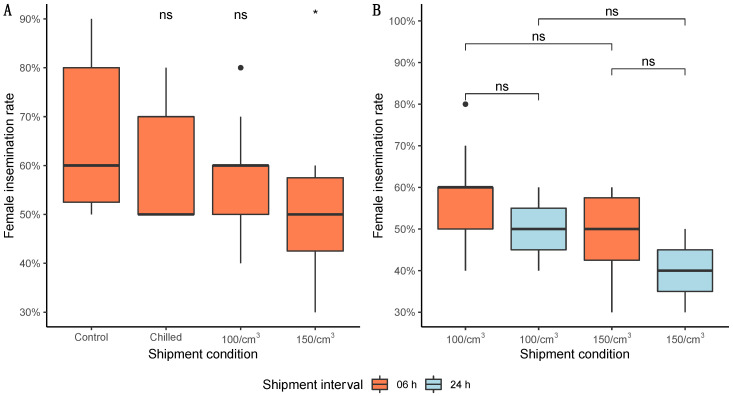
Mean insemination rate *Aedes aegypti* females after mating with males exposed to different shipment conditions: (**A**) shipment period of 6 h with comparisons always using the control group as reference; (**B**) comparison of 6 and 24 h for both densities (N = 80 females/group; 8 males/group, 240 spermathecae/group). Significance values goes as ‘*’ < 0.05, ‘ns’ < 0.1. Black dots correspond to potential outliers among the samples.

**Figure 5 insects-13-00871-f005:**
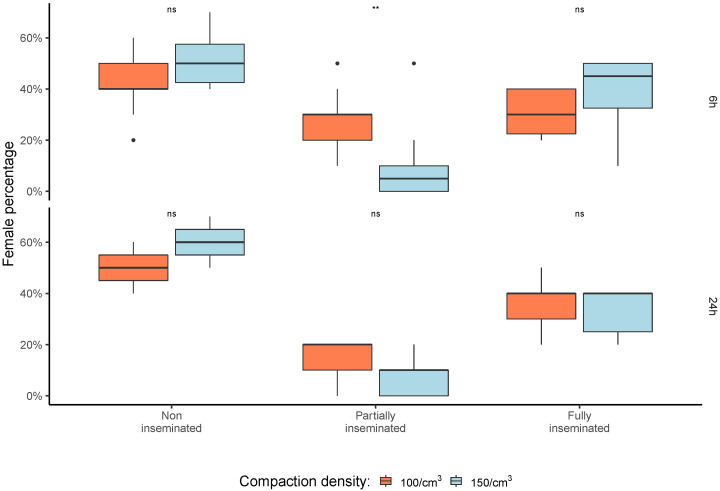
Insemination level of *Aedes aegypti* females after mating with exposed males at two different compaction densities and two shipment periods (N = 80 females/group; 8 males/group) (Wilcoxon–Mann–Whitney, *p* < 0.05). Significance values goes as ‘**’ < 0.01, ‘ns’ < 0.1. Black dots correspond to potential outliers among the samples.

**Figure 6 insects-13-00871-f006:**
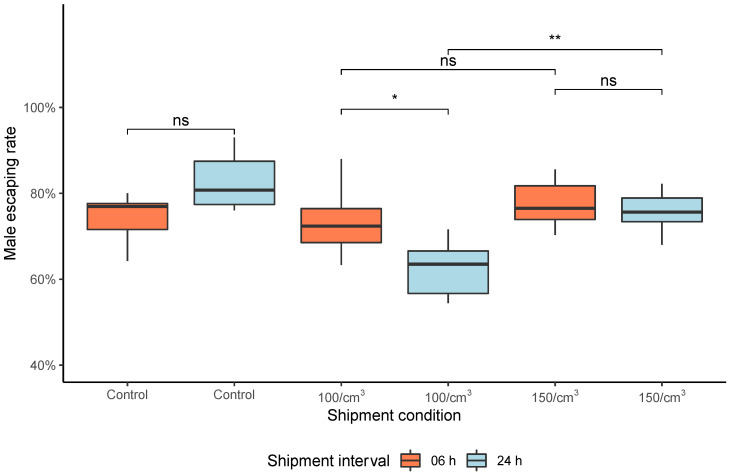
Escape rates of male *Aedes* mosquitoes after knockout (4 °C for 60 min), compacted (100 and 150 males/cm^3^), and transported (for 6 and 24 h at 10 °C). Significant differences between treatment groups are indicated by the *p*-value level. The boxplot shows the median and upper and lower quartiles. Significance values goes as ‘**’ < 0.01, ‘*’ < 0.05, ‘ns’ < 0.1.

**Figure 7 insects-13-00871-f007:**
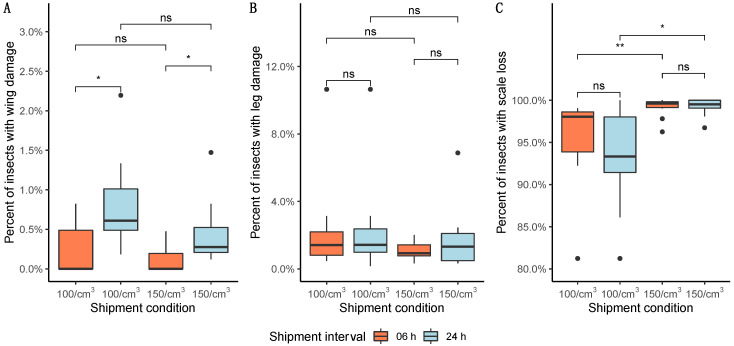
Physical damage after compaction (100 and 150 males/cm^3^) and shipment interval (06 and 24 h) at 10 °C: (**A**) wing damage; (**B**) leg damage; (**C**) scale loss. Significant differences between treatment groups are indicated by the *p*-value level. The boxplot shows the median and upper and lower quartiles. Significance values goes as ‘**’ < 0.01, ‘*’ < 0.05, ‘ns’ < 0.1. Black dots correspond to potential outliers among the samples.

## Data Availability

The data presented in this study are available in Appendix A.

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
