# Peer review of "Exploring Conditions for Handling Packing and Shipping Aedes aegypti Males to Support an SIT Field Project in Brazil"

_insects, 2022, doi:10.3390/insects13100871_

Round 1

Reviewer 1 Report

General comment:

1.           The impact of chilling, compacting, and transporting procedures on male quality are key points in SIT program. Therefore, the content of this manuscript is of practical value for reference. But the biggest drawback of this study is the absence of irradiation treatment for male mosquitoes. it may make the references that the author cited irrelevantly. Because male mosquitoes may have different physiological changes from normal male mosquitoes after exposure to radiation, the subsequent processing conditions must be different.  This is a fundamental question that should be pondered by all who study mosquito SIT.

2.           Apart from the shortcoming, the authors showed comprehensive results on how the males should be packed and transported from the rearing/irradiation facility to the release site. Experimental designs are well-executed and the results answered the aims accordingly.

3.           One of the purposes of the draft is to find out an appropriate condition to apply to SIT mosquito program in Brazil. Thus, all physic conditions in the laboratory and the simulating shipment should be detail documented in sections 2.2 and 2.3 and discussed in the discussion section. For example, 1) location of the incubator, because you took out mosquitoes from the chilling condition, the mosquitoes would be heated by environment temperature. Thus, we cannot know if the mosquitoes face two temperature changes from chilling conditions to the environment, and then transfer to the mosquito room or just only one time from chilling to mosquito room. These factors should be described. 2) the material of the container (unknown) in the chilling experiment might be different from the packing experiment (glass), and the different heat conduction of the materials might affect the chilling effectiveness for testing mosquitoes.

4.           The word “shipping/shipment” may be inappropriate in the shown experiments, the experiments seem to test the packing and chilling condition using the containers of field SIT application without shipping, but the procedure of shipment will face factors like car shaking, temperature changes, air pressure changes…etc. Here, a multi-way Chi-square test is suggested for understanding the relationships among packing, chilling, and other factors.

5.           Since this study only tested healthy male mosquitoes that were not exposed to radiation, the results, although informative, should discuss whether the condition of the process described in this manuscript still applies once the male mosquitoes have been exposed to radiation.

6.           Things that may need to improve the detailed information regarding the control groups. Thorough reasoning of why the particular events may happen and their impact on the handling, packing, and release strategy. These are all valuable information for to process of male mosquitoes. I recommend minor revision prior to acceptance.

7.           Finally, since the male mosquitoes tested in this article have not been exposed to radiation, because irradiation treatment may give you completely different results/stories it is recommended to modify the title to "Exploring conditions for packing and shipping Aedes aegypti males as a reference to SIT field project in Brazil". Or the author can set a more suitable title by themselves.

Detailed comments and questions:

Line 79-80: please check the accuracy of the references. Tur et al., 2021 actually do not show the successful project of reducing the target mosquitoes. The article tells the readers their readiness to conduct the operating procedures and quality control prior to the open releases.

Line 87-89: It would be great if you can put a reference(s) to support the statement.

Line 93-101, the authors did not provide more advanced experiments than the references 35 to 40, especially using the mosquitoes without irradiation. However, the appropriate conditions for different strains of mosquitoes using different transport ways should be certainly tested. If it's possible, rewrite the final section of the introduction for fitting the purpose of the authors' experiment.

Line 108-109: If this is the case (Irradiation treatment was absent), your cited references have become irrelevant.

Line 117: I tried to search “incubator (Labom; model SP-227/364)” in the search engine and found nothing. Please specify the ID or simply put a picture of the incubator for better reproducibility.

Line 156-161: Please specify the number of specimens you were using.

Line 164-165: this sentence is hard to read. Please rephrase. Maybe: We determined the number of inseminated females after 15 hours caged together with a single male from the treatments tested. The number of females for this experiment was limited to 10 individuals.  

Line 218-220: I couldn’t find the impact of age (24 hours later) in MM section. Please add a line to describe it.

Figure 1. Control of sub-figure A and B should be different which will lead to misunderstanding of the figure for readers, please give the right control label in sub-figure A and B.

Figure 2: a bit confusing when in the text you mentioned 24 hours after chilling treatment, but in the figure you put hours after emergence. Better to have a consistent time parameter.

Figure S1B: please decide on the decimals in your P value.

Line 228: “seem not to have an effect” does not sound scientific. You have proven it by doing statistical analysis.

Figure 4A: what do P values tell the readers?

Figure 3 A and B, 4A: You should consider describing the control and chilled treatment in MM.  In my understanding, control did not receive any treatment, neither packing nor shipping in chilled condition. How about chilled? Were they packed? Maybe it would be easier for the reader to understand the name of axis of chilled-100/cm3 and chilled-150 cm3.

Line 333-345: the exact time required to completely knock down the males should be addressed in this section to answer: why do we need 60 min if 15 min, for example, at 4 oC is enough to make the males fall off the container walls.

Line 346-355: A deep explanation of why 24 hours recovery time after the treatments improved the escape rate should be added. It actually reveals an insight into the release strategy, such as the need for a recovery room at the targeted release site.

Line 356-363: This paragraph seems irrelevant to this study’s objectives.

Line 372: Yes, it is actually the transportation time that matters, not the long distance of the targeted site as mentioned in line 23.

Line 374-375: this actually the opposite of what was written in 356-363. The escape rate should be more sensitive to detect the drawbacks, considering the highest compaction resulted in high mortality.

Line 376-386: Does it related to oxygen availability?

Line 432-433: It is disappointing and puzzling when the authors do not recommend continuing the study by adding an irradiation variable.

Author Response

We appreciate all reviewer's comments and suggestions, trying to address the best we could. Please see the attachment.

Reviewer 2 Report

A good paper with a VERY limited interest

Author Response

We appreciate the reviewer's comment, despite the unnecessary ending, which it was not clear.

Reviewer 3 Report

Overall, I think the paper is written in a simple and easy-to-understand manner. I have only the following very minor comments/suggestions.

Simple summary

Please change ‘the outcomes’ to ‘these outcomes’.

Abstract

Line 20: alternative tool for conventional mosquito control methods….

Introduction

No comments.

Materials and methods

No comments.

Results

No comments.

Discussion

I think the authors should include discussion about how these traits may possibly vary in irradiated mosquitoes.

Author Response

We appreciate the reviewer's comment and suggestions. Please see attachment.
